# Modular Combinatorial DNA Assembly of Group B Streptococcus Capsular Polysaccharide Biosynthesis Pathways to Expediate the Production of Novel Glycoconjugate Vaccines

**DOI:** 10.3390/vaccines13030279

**Published:** 2025-03-06

**Authors:** Mark A. Harrison, Elizabeth Atkins, Alexandra Faulds-Pain, John T. Heap, Brendan W. Wren, Ian J. Passmore

**Affiliations:** 1Department of Infection Biology, London School of Hygiene & Tropical Medicine, London WC1E 7HT, UKbrendan.wren@lshtm.ac.uk (B.W.W.); 2School of Life Sciences, University of Nottingham, Nottingham NG7 2RD, UKjohn.heap@nottingham.ac.uk (J.T.H.)

**Keywords:** glycobiology, glycoconjugates, vaccines, DNA assembly, group B Streptococcus, start-stop assembly

## Abstract

Background/objectives: *Streptococcus agalactiae* (or Group B Streptococcus, GBS) is a major cause of neonatal meningitis globally. There are 10 serotypes of GBS, which are distinguished by their capsular polysaccharide (CPS) structure, with serotypes Ia, Ib, II, III, IV and V responsible for up to 99% of infections. Currently, there are no licensed vaccines against GBS. The most developed candidates are glycoconjugate vaccines, which can be highly effective but are also expensive to produce by existing approaches and unaffordable for many parts of the world. Biosynthesis of recombinant glycans and glycoconjugates in tractable strains of bacteria offers a low-cost alternative approach to current chemical conjugation methods. Methods: In this study, we apply combinatorial hierarchical DNA assembly to the heterologous biosynthesis of GBS III, IV and V CPSs in *E. coli*. Each gene was removed from its native regulation, paired with synthetic regulatory elements and rebuilt from the bottom up to generate libraries of reconstituted pathways. These pathways were screened for glycan biosynthesis using serotype-specific antisera. Results: We identified several configurations that successfully biosynthesised the GBS CPSs. Furthermore, we exploited the conserved nature of the GBS CPS biosynthesis loci and the flexibility of modular DNA assembly by constructing hybrid pathways from a minimal pool of glycosyltransferase genes. We show that transferase genes with homologous function can be used interchangeably between pathways, obviating the need to clone a complete locus for each new CPS assembly. Conclusions: In conclusion, we report the first demonstration of heterologous GBS CPS IV and V biosynthesis in *E. coli*, a key milestone towards the development of low-cost recombinant multivalent GBS glycoconjugate vaccines.

## 1. Introduction

Bacteria are ideal hosts for the heterologous biosynthesis of complex glycan structures such as *N*- and *O*-linked glycoproteins, capsular polysaccharides and lipooligosaccharides. Recombinant glycans can be linked in vivo to other macromolecules such as proteins and lipids to generate glycoconjugate vaccines, as well as diagnostics and therapeutics [1,2,3,4,5,6]. This technology, known as protein glycan coupling technology (PGCT) or bioconjugation, offers great flexibility to the custom design and biosynthesis of a wide range of glycan structures from diverse bacterial species [7,8]. Central to this process is the ability to faithfully reconstitute a glycan biosynthesis pathway from the organism of interest in a non-native tractable bacterium, such as *Escherichia coli*. Typically, this has been attempted by cloning an unmodified locus en bloc into a low-copy plasmid and transforming it into the new host. While this sometimes results in a functional glycan biosynthesis pathway, this process is technically challenging, and glycan yields can be lower than desired. In recent years, advances in engineering biology have precipitated a paradigm shift in approaches to the heterologous expression of multi-gene systems [9]. Principles such as pathway refactoring (deconstructing multi-gene pathways and rebuilding them in a modular format) and DNA assembly by combinatorial construction have provided convenient platforms for building and optimising such complex multi-gene pathways, resulting in a marked improvement in product yield [10,11,12]. We recently applied combinatorial assembly and pathway refactoring to the heterologous biosynthesis of the *N*-glycosylation pathway from *Campylobacter jejuni* in *E. coli* [13]. In this study, we identified pathway clone variants that outperformed the glycan and glycoconjugate production of the native unmodified locus, suggesting that this approach could readily be applied to the biosynthesis of diverse recombinant glycostructures [13].

*Streptococcus agalactiae*, commonly referred to as Group B Streptococcus (GBS), is a major cause of neonatal morbidity and mortality globally and causes an estimated 100,000 deaths per year [14]. GBS is a commensal of the intestinal tract and the vagina, and an estimated 19.7 million pregnant women are colonised worldwide [15]. GBS can cause opportunistic infections in neonates, postpartum women and individuals with immune impairment, and can lead to life-threatening meningitis, sepsis and pneumonia. Two major syndromes of invasive neonatal disease have been classified by age of onset: early onset of infection occurring within 0 to 6 days of life and late onset from days 7–90. Maternal colonisation is the leading risk factor for neonatal exposure and infection. During the 2015 WHO Product Development for Vaccines Advisory Committee meeting, GBS was identified as a top priority for vaccine development for maternal immunisation because of the major public health burden posed by GBS in low- and middle-income countries [16,17]. A GBS vaccine would protect newborns from early and late onset GBS disease through the transfer of transplacental maternal antibodies. In addition, the emergence of predominant serotypes and increased antibiotic resistance further highlights the need for flexible strategies in vaccine development.

Opsonophagocytosis is the main mechanism for the immune system to clear GBS infection, and a recent Phase II study demonstrated that the vaccination of pregnant women with a trivalent glycoconjugate vaccine induced the formation of serotype-specific opsonophagocytic antibodies [18]. GBS can also cause invasive infection in non-pregnant adults—particularly the elderly and immunocompromised—with the burden of infection rising, which has led to calls for vaccination within this population [19] Additionally, GBS outbreaks in aquaculture, such as the farming of Tilapia, cause significant economic damage. As such, efforts have been made to develop a vaccine for this industry (reviewed by Maulu et al. [20]). 

There are 10 serotypes of GBS, and serotypes Ia, Ib, II, III, IV and V are estimated to cause up to 99% of infections [21]. Several vaccines are currently in development, including Pfizer’s hexavalent GBS6, a capsule-based glycoconjugate vaccine which is in phase II clinical trials [22]. This vaccine is generated by purifying the capsule glycan from the six major circulating GBS serotypes and their chemical conjugation to modified diphtheria toxin, CRM197. The use of chemically made glycoconjugate vaccines has been a major success for public health; however, the manufacture of these vaccines is an expensive and complex process, leading to vaccines that are unaffordable for large parts of the world. In recent years, bioconjugation is increasingly being used for the generation of glycoconjugate vaccines, such as for Shigellosis and extraintestinal *E. coli* [23,24]. Bioconjugation offers numerous advantages over chemical conjugation, including greater flexibility in design and the potential for significantly lower manufacturing costs [7,8]. A recent study by Duke et al. demonstrated the recombinant biosynthesis of the GBS capsular polysaccharide (CPS) of serotypes Ia, Ib and III in *E. coli* and their bioconjugation to *Pseudomonas aeruginosa* exotoxin A (ExoA), which elicited a functional immune response in mice [25]. This study is a significant step towards the application of PGCT and bioconjugation as an alternative to chemically conjugated vaccines for the prevention of GBS infections.

While each GBS CPS structure is antigenically distinct, their repeat unit structures are remarkably similar in their monosaccharide compositions. Evidence suggests that the acquisition of individual glycosyltransferase genes has occurred through horizontal gene transfer rather than stepwise mutagenesis [26]. A number of biosynthesis and transferase genes are common to all CPS pathways (*neuB*, *neuC*, *neuD*, *neuA*, *cpsE*, *cpsF* and *cpsL*), while others are shared between two or more serotypes (e.g., *cpsH* in CPSII, III and VI; *cpsI* in Ia, Ib, II and III) [26]. From an engineering perspective, this presents an opportunity towards designing optimal recombinant glycan biosynthetic pathways. Core genes (common to all CPS pathways) can be combined with accessory genes (unique to each CPS pathway) to generate full pathways with hybrid configuration. This approach obviates the need to clone a new complete genetic locus for each CPS pathway. Instead, single-gene or multi-gene modules can be selected based on known (or predicted) activity and screened for optimal performance, and the best-performing configurations can be repurposed in new pathway assemblies.

In this study, we applied the principles of combinatorial hierarchical DNA assembly to the heterologous biosynthesis of GBS capsule glycans. We exploited the conserved nature of the GBS CPS loci and the modularity of hierarchical assembly to construct hybrid glycan biosynthesis pathways using genes from multiple pathways. Using this method, we demonstrate the biosynthesis of the GBS CPS III, IV and V in *E. coli*, a key step towards production of glycoconjugate vaccines through bioconjugation.

## 2. Materials and Methods

### 2.1. Strains and Plasmids

All strains and plasmids used in this study are listed in Table 1 and Table 2, respectively.

*E. coli* cells were cultured on Luria–Bertani (LB) agar and LB broth at 37 °C. Antibiotics were added where appropriate at the following concentrations: ampicillin 100 µg/mL, tetracycline 20 µg/mL, kanamycin 50 µg/mL and chloramphenicol 30 µg/mL.

### 2.2. Construction of GBS CPS Biosynthesis Pathways

Individual parts, expression units and biosynthesis pathways were constructed by combinatorial hierarchical Start-Stop Assembly, as described by Taylor et al. [9,28]. Synthetic DNA sequences were synthesised by Integrated DNA Technologies (IDT) and are listed in Appendix A. All sequences were codon optimised for expression in *E. coli* K12, with restriction sites for SapI, BsaI and BbsI removed using IDT’s codon optimisation tool. Destination plasmids for the coding sequences are shown in Figure 1. Level 0 and 1 constructs were verified by Sanger sequencing carried out by Eurofins Genomics. The oligonucleotides used are listed in Table 3.

### 2.3. Detection of Glycan Expression by Dot Blot Analysis

Following transformation of DH10β with Level 3 plasmids, two approaches were initially used to transform W3110 (Appendix A). Approach one used individual colonies of DH10β which were grown overnight, and plasmid DNA was isolated by miniprep. Plasmid DNA was sequenced by Plasmidsaurus Inc, and plasmids that contained correctly assembled pathways were transformed into W3110. In approach two, after transformation of DH10β, white colonies were patch-plated onto a fresh LB agar plate. From this plate, all colonies were collected, and plasmid DNA was isolated by miniprep to generate a library of constructs, with this library used to transform W3110. Transformation of W3110 was carried out as follows: W3110 was grown overnight in LB broth, and 200 µL was inoculated into 10 mL fresh LB and grown to exponential phase (approximately OD_590_ 0.6). Cells were pelleted at 3234× *g* and washed in 5 mL of 10% (*v*/*v*) glycerol followed by two further washes with 2 mL 10% glycerol. After the final wash, cells were resuspended in 150 µL of 10% glycerol and plasmid DNA was added prior to electroporation. Cells were recovered in Super Optimal broth with Catabolite repression media (SOC) for 1 h at 37 °C with shaking before growth on LB agar with chloramphenicol.

Individual colonies were picked and grown in 200 µL LB chloramphenicol in 96-well polystyrene plates, shaking at 37 °C at 600 rpm overnight. Cells were pelleted by centrifugation (1683× *g*), washed three times with 300 µL PBS and resuspended in 150 µL PBS. Two microlitres of cell suspension was spotted onto nitrocellulose membrane and air-dried for 30 min. Membranes were washed three times for 10 min using PBS with 0.1% *v*/*v* Tween20 (PBST). After the final wash, a 1:1000 dilution of the appropriate anti-sera (all anti-sera from Statens Serum Institut, Denmark) in PBST was added and the membranes were incubated for one hour. Membranes were washed three times for 5 min with PBST before incubation with the secondary antibody IRDye800CW Goat-Rabbit (LI-COR biosciences, Lincoln, NE, USA) at a dilution of 1:10,000 in PBST for 45 min. A final set of three 5 min washes in PBST was performed before detection of the fluorescent signal by the Odyssey LI-COR system (LI-COR biosciences, Lincoln, NE, USA).

### 2.4. Western Blot Analysis

Cells were grown in 5 mL LB broth overnight at 37 °C with shaking. Cells were normalised to the same cell density (OD_590_ = 1), pelleted by centrifugation (8000× *g*), the supernatant discarded and the pellet re-suspended in 100 µL 1× LDS buffer (Thermo Fisher Scientific, Carlsbad, CA, USA). Then, 1.5 µL of 800 U / mL Proteinase K (New England Biolabs) was added and the samples were incubated at 55 °C for 1 h before heating at 100 °C for 10 min. Cell contents were resolved by SDS-PAGE (4–12% Bis-Tris SDS-PAGE gel) and transferred to a nitrocellulose membrane using an iBlot 2 dry blotting system (Thermo Fisher Scientific, Carlsbad, CA, USA). Glycan detection was carried out as above for the dot blots.

## 3. Results

### 3.1. Heterologous Biosynthesis of GBS CPS III

A recent study by Duke et al. demonstrated the feasibility of heterologous biosynthesis of the GBS III glycan in *E. coli* cells [25]. Therefore, we selected this locus as the first to assemble using a combinatorial hierarchical approach. The genetic locus for GBS III contains 12 genes (excluding transcriptional regulators), which encode glycosyltransferases (*cps3E*, *cps3G*, *cps3I*, *cps3J*, *cps3K*), a glycosyltransferase enhancer (*cps3F*), a flippase (*csp3L*), a polymerase (*cps3H*) and sialic acid metabolic enzymes (*neuB*, *neuC*, *neuA*, *neuD*) (Figure 1). Cloning of large genetic loci such as those required for biosynthesis of the *cps* locus is challenging for several reasons, including difficulties in primer design, amplification of long contiguous PCR fragments, incompatibility of sequences with *E. coli* carrier strains and difficulties identifying regulatory elements required for expression. Hierarchical assembly platforms overcome many of these challenges by deconstructing pathways into individual transcriptional units and building them back into full pathways in discrete steps.

Each coding DNA sequence (CDS) in the GBS III locus was chemically synthesised and cloned into the Level 0 Start-Stop Assembly storage vector, pStA0. These were subsequently assembled into Level 1 single-gene transcription units using a standard set of two promoters (P4 and P5), six ribosome binding sites (RBSs) (R1 to R6) and a single transcriptional terminator (T1). Equimolar mixtures of the promoter, RBS, CDS and terminator parts were combined in a single-pot assembly reaction to generate a library of Level 1 plasmids with a possible total of 12 different promoter–RBS combinations (and hence expression strengths) for each CDS (Figure 1A). Successfully assembled plasmids were evaluated by blue/white screening and transformants were used to prepare plasmid DNA. Correctly assembled plasmids were identified for every CDS in the pathway, under both P4 and P5 promoters, except for *neuD*, which encodes a sialic acid *O*-acetylase. However, it has previously been shown that sialic acid *O*-acetylation is unnecessary for a functional immune response [13]. Similarly, Duke et al. showed that GBS III sialic acid was not *O*-acetylated when produced recombinantly in *E. coli* and that mice immunised with the resulting glycoconjugates still raised functional opsonophagocytic antibodies against wild type GBS III [25].

These Level 1 transcription units were subsequently used to build one five-gene (*cpsE*, *cpsF, cpsG*, *cpsH* and *cpsI*), and two three-gene pathways (*cpsJ*, *cpsK*, *cpsL* and *neuB*, *neuC*, *neuA*) in Level 2 vectors, pStA2 (Figure 1A). Transformants were screened by sequencing. A proportion of transformants contained mutations in CDSs or were misassembled. Only correctly assembled plasmids were used for Level 3 assemblies, carrying all the required genes for synthesis of the CPS. A total of six unique pathways for *cpsEFGHI*, four unique pathways for *cpsJKL* and five unique pathways for *neuBCA* were combined to generate a full pathway capable of synthesising the GBS CPSIII locus, representing a final library size of 120 possible combinations.

Plasmids were transformed into the *E. coli* cloning strain DH10β due to its high transformation efficiency and capacity for blue/white screening. However, this *E. coli* strain has mutations in *galE* and *galK* and is impaired in its ability to biosynthesise UDP-gal, a constituent monosaccharide in all GBS CPS structures. Strains such as W3110, which can synthesise UDP-gal, are better suited for screening GBS surface glycans but have lower transformation efficiency, which creates a bottleneck in screening large combinatorial libraries. Furthermore, we speculated that misassembled plasmids containing incomplete pathways would be preferentially transformed into W3110 due to their smaller size and lower metabolic burden on the cells. Considering this, two parallel approaches to screening were adopted. In approach one, full biosynthesis pathways were assembled and transformed into DH10β, single colony clones were selected and sequenced, and those that had assembled correctly (6 unique clones) were directly transformed into W3110 (Appendix A). In approach two, full pathways were transformed into DH10β and white colony transformants were pooled into a single plasmid DNA preparation, which was used to transform W3110 as a library (Appendix A). In both approaches, individual W3110 clones were cultured in 96-well plates, washed to remove culture medium and cell suspensions were transferred to a nitrocellulose membrane. Biosynthesis of glycan was assessed by probing cell spots with CPS III-specific antiserum. For the first approach, 91 of 96 clones displayed cross reactivity with the antiserum, which indicated biosynthesis of GBS CPS III on the cell surface (Figure 2A). For the second screening approach using a library of clones, 78 of 96 clones displayed cross-reactivity (Appendix A).

Between the two approaches, eight clones were selected (six from approach 1 (CPSIII.1 to CPSIII.6) and two from approach 2 (CPSIII.7&8)), cultured, normalised to the same OD (OD_590_ = 1), lysed, cell contents were resolved by SDS-PAGE and glycan detected by immunoblot. Highly polymerised glycan was observed in all pathway variants (Figure 2B).

A five-gene pathway containing the initial genes of the GBS CPS III (*cpsEFGHI*) biosynthesis locus was assembled, transformed into W3110 and included as a control. Surprisingly, cell extracts from this pathway also cross-reacted with CPS III antiserum, suggesting the formation of partial CPS III. However, these two glycan structures could be differentiated by probing with the lectin Mal I, which binds gal(β-1,4)-glcNAc, a structure that is predicted to be catalysed by the transferase CpsJ and is only present in the full pathway (Appendix A).

### 3.2. Heterologous Biosynthesis of GBS CPS IV and CPS V

The genetic loci for capsular polysaccharide biosynthesis of GBS serotypes IV and V contain 12 and 13 genes, respectively (excluding the regulatory genes and *neuD*). Six genes are common to the GBS III, GBS IV and GBS V pathways (*cpsE*, *cpsF*, *cpsL*, *neuB*, *neuC* and *neuA*). There are six additional genes in the GBS IV pathway (*cps4G*, *cps4H*, *cps4I*, *cps4J*, *cps4M* and *cps4K*), two of which are also present in the GBS V pathway (*cpsG* and *cpsH*). GBS V contains four additional genes absent in either GBS III and GBS IV (*cps5I*, *cps5J*, *cps5M* and *cps5O*) (Figure 1B). Theoretically, only 21 unique genes are required for synthesis of the three glycan structures.

CPS biosynthesis pathways for GBS IV and GBS V were designed as indicated in Figure 1B. As for the GBS III pathway, two promoters, six RBSs and a single transcriptional terminator were used to generate single-gene transcription units. Genes encoding enzymes with orthologous functions in multiple pathways were cloned into the same Level 1 plasmids where possible. These Level 1 plasmids were used to generate intermediate multi-gene pathways in Level 2 vectors, reflecting the concept of placing expression units into Level 2 groups encoding different functional components to be optimised or reused independently [28]. Transformants were screened, sequenced and, as before, only those pathways that had correctly assembled were used to generate full CPS pathways. Four unique plasmids for *cps4EFGHI* and three plasmids for *cps4JKL* were combined with five plasmids for *neuBCA* to generate the full GBS IV pathway, representing a total library size of 60 unique combinations. For GBS V, five unique plasmids for *cps5EFGHI* and six for *cps5JKLMO* were combined with the five *neuBCA* plasmids to generate the final GBS V pathway, representing a total library size of 150 unique combinations. We opted to utilise approach one for generating W3110 expressing the *cps* loci as for GBS III we saw no visible improvement in quantity of glycan expressed or in polymer length when approach two was used. Plasmids were transformed into *E. coli* W3110 cells and production of CPS was assessed by dot blot using serotype-specific antiserum. For both GBS IV and V, four plasmids were transformed into W3110 and screened initially by dot blot. For GBS IV, clones from each pathway displayed cross-reactivity with antiserum, suggesting the biosynthesis of surface-exposed glycan (Figure 3A). For GBS V, cross reactivity with antisera was observed in 3 out of 4 of the pathways (Figure 4A) with no expression with any clone transformed with CPSV.3. However, two of the three pathways yielded only a single cross-reactive colony (CPSV.1 and CPSV.4), which indicates that these configurations may have been toxic and accumulated mutations (Figure 4A). Clones from both assemblies were selected and further assessed for glycan production by SDS-PAGE and immunoblot. Highly polymerised glycan was observed in all GBS IV and GBS V biosynthesis pathways, indicating that these hybrid loci were fully functional (Figure 3B and Figure 4B).

Taken together, these data demonstrate the advantages of combinatorial assembly for construction of complex glycan biosynthesis pathways and recombinant production of therapeutic glycans. Construction of hybrid pathways obviates the need to clone a complete complement of genes for each new CPS assembly. Rather, each gene can be selected based on its function and orthologous genes can be utilised across multiple pathways.

## 4. Discussion

In this study, we employed a combinatorial hierarchical assembly strategy to efficiently assemble biosynthetic pathways for the production of GBS CPSs recombinantly in *E. coli* to circumvent the challenges associated with en bloc cloning. Using this approach, we assembled and synthesised three GBS capsular polysaccharides in *E. coli,* including two clinically important serotypes which to date have not been produced recombinantly (GBS IV and GBS V). Furthermore, we exploited the modularity of combinatorial assembly and the conserved nature of GBS CPS structures to assemble hybrid pathways from a minimal pool of CDSs.

A number of combinatorial DNA assembly techniques have been developed in recent years for the efficient assembly and improved biosynthesis of non-native metabolic pathways in a range of model host organisms. For example, the COMPASS [29] and VEGAS [30] systems have been demonstrated in *Saccharomyces cerevisiae*. In bacteria, the JUMP platform is similar to Start-Stop Assembly in that both use type IIS restriction enzymes for generating discrete transcriptional units for individual genes, which are then combined to make multi-gene plasmids [31]. Another system is the ePathOptimize platform, which makes use of a library of isopropyl β-D-1-thiogalactopyranoside (IPTG)-inducible mutant T7 promoters, and has been used to improve fermentation yields of violacein in *E. coli* [32]. The expansion in the development of such techniques highlights the challenges associated with heterologous biosynthesis of non-native pathways (and the potential merits of adopting these techniques). Unlike proteins, synthesis of glycans is not templated, but is the product of the concerted action of multiple enzymes within the metabolic context of the host organism. Compounding on this, several different enzyme classes are involved, including nucleotide-linked sugar metabolism enzymes, glycosyltransferases, polymerases and ligases, which typically operate in different cellular compartments. Any attempt to rationally modify bacterial cells or design optimal glycan biosynthesis loci requires substantial a priori knowledge of the endogenous and incoming pathways and how they may interact. Therefore, combinatorial DNA assembly coupled with non-biased screening methods, such as that described in this study, are well-suited to overcoming these challenges towards more efficient production of recombinant glycans and glycoconjugates.

Compared with conventional cloning techniques, combinatorial assembly approaches facilitate the identification of components that are toxic or cause issues in pathway construction [28]. Additionally, with its standard sets of promoters and RBSs, Start-Stop Assembly allows for the straightforward manipulation of the expression of individual genes to reduce their toxicity or to express genes to a higher level to increase the biosynthesis of the final product. In this study, we observed several genes that had acquired mutations when assembled as multi-gene pathways. Using such a stepwise assembly approach, these mutations and broken pathways could be readily identified and excluded from subsequent assembly steps. However, while this resulted in functional CPS biosynthesis, it came at the cost of a reduced design space and a smaller pool of clones to screen. Whereas the potential library size for the GBS III locus was (2 × 6)^13^ = 1.06 × 10^14^, only eight unique combinations were tested. Therefore, there are likely to be further pathway configurations with improved (or reduced) glycan yield, as was observed when using modular assembly for expression of the *C. jejuni* heptasaccharide in *E. coli* [13]. The identification of which individual genes and gene combinations place a metabolic burden on *E. coli* could lead the way for further optimisation of the design of expression libraries.

There is a current imperative to develop a low-cost GBS vaccine. A recent study highlighted the high global burden of GBS disease, which causes an estimated 392,000 cases of invasive GBS, 91,000 deaths, 46,000 stillbirths, 40,000 cases of neurodevelopmental impairment and 518,000 preterm births annually [33]. The study concluded that an effective GBS vaccine could reduce disease in the mother, foetus and infant, and provides a compelling rationale for vaccine development for all stakeholders [33]. Currently, preventative measures for GBS infection in neonates are limited to screening programs to identify colonisation of the mother. At-risk mothers are given intrapartum antibiotic prophylaxis (IAP), which is effective at preventing early-onset invasive GBS disease. However, screening programmes and administration of intravenous IAP are not universally available, particularly in low- and middle-income countries [34]. Furthermore, such prevention strategies are vulnerable to the emergence of antimicrobial resistance in GBS, such as the rise of erythromycin and clindamycin resistance seen in GBS isolates [35]. The introduction of an effective vaccine against Group B Streptococcus would be highly impactful for neonates and their parents, as well as non-pregnant adults who suffer infection. A large part of GBS vaccine development has focused on glycoconjugate vaccines utilising the CPS conjugated to a carrier protein, with the efficacy of glycoconjugates shown in both preclinical and clinical trials [18,36,37]. However, bioconjugation offers a promising alternative strategy through reduced and simplified manufacture and therefore reduced overall vaccine costs [7,8]. Duke et al. used the *Acinetobacter baylyi* oligosaccharyltransferase PglS to bioconjugate the CPS of serotypes Ia, Ib and III to a carrier protein and successfully demonstrated immunogenicity of their candidate vaccines against these serotypes [25] but serotypes II, IV and V would ideally need to also be included to cover the vast majority of disease-causing strains [21]. Therefore, our work demonstrating the heterologous biosynthesis of serotypes IV and V CPSs is a significant step towards a universal bioconjugate hexavalent vaccine. Validation of a functional immune response to these glycans and any glycoconjugates that include them is essential. Therefore, future research should include a combination of in vitro and in vivo assays, such as opsonophagocytic killing (OPK), and a challenge model of infection following vaccination.

In recent years, GBS has become an increasing problem in global aquaculture, causing disease outbreaks in Tilapia (*Oreochromis* spp.) and resulting in food chain supply disruptions and economic loss, particularly in low-and-middle income countries [20]. Development of a low-cost, easy-to-administer vaccine is a promising strategy for preventing these disease outbreaks and curtailing the use of antibiotics in aquaculture. Current vaccines in development include inactivated bacterial cells (typically by formalin inactivation), live-attenuated strains and subunit vaccines [38,39,40,41,42]. In most instances the vaccine antigens are derived directly from GBS itself, which creates a comparable bottleneck to the development of affordable human vaccines. In this study, we present an alternative method for generating these antigens through their cell surface display in *E. coli*. Using *E. coli* as a delivery vector (administered as killed or live attenuated cells) for these antigens is an appealing alternative to GBS-based vaccine delivery, due to its simpler and safer processes for biomanufacture.

## 5. Conclusions

In summary, we have demonstrated the flexibility of modular combinatorial assembly towards the biosynthesis of GBS polysaccharides from a minimal number of genes. Furthermore, this is the first time GBS IV and V CPSs have been synthesised recombinantly, demonstrating the feasibility of this method for future use in vaccine design.

## Figures and Tables

**Figure 1 vaccines-13-00279-f001:**
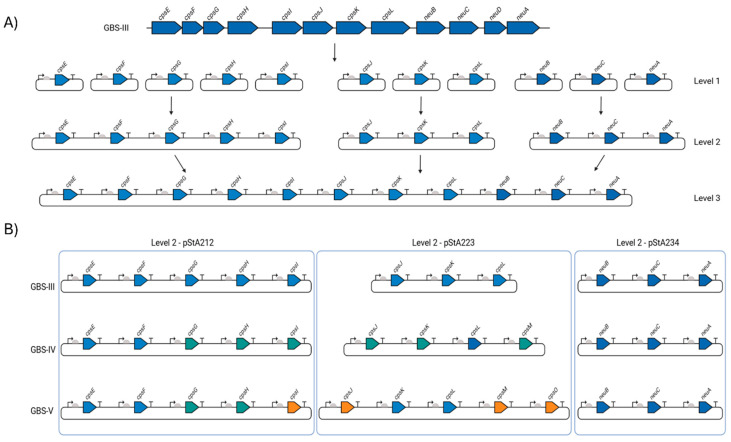
Start-Stop Assembly of GBS biosynthesis loci. (**A**) Schematic of Start-Stop Assembly [9] with GBS III as an exemplar. Each CDS from the GBS III capsular polysaccharide locus was cloned and assembled into single-gene transcriptional units with defined sets of promoter (arrows), RBS (semi-circles) and transcriptional terminator (T) parts (Level 1). Single transcriptional units were then combined into three multi-gene units (Level 2) and complete biosynthesis pathways (Level 3). (**B**) Configuration of GBS III, GBS IV and GBS V CPS biosynthesis pathways. Blue arrows are genes encoded by the GBS III *cps* locus and which are common to GBS IV and GBS V. Turquoise arrows denote genes encoded by GBS IV and which are common to GBS V. Orange arrows are those encoded only by GBS V. Level 2 plasmid configurations are denoted and grouped within the blue boxes. Figure created with Biorender.com (URL accessed 17 February 2025).

**Figure 2 vaccines-13-00279-f002:**
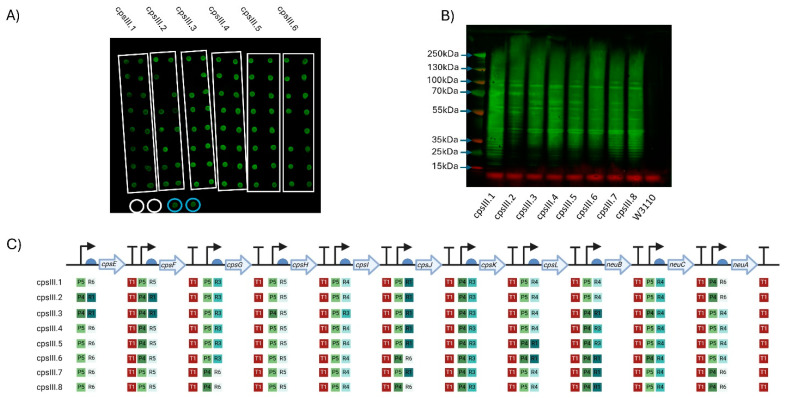
Expression of GBS serotype III CPS in W3110. (**A**) Dot blot for screening expression of CPS III. Clones of six plasmids (cpsIII.1–cpsIII.6) were screened, with colonies for each demarcated by the white boxes. Circled in white are biological replicates of W3110 only, used as a negative control, circled in blue are positive controls expressing *cpsEFGHI* only. Detection was carried out using anti-GBS III antisera. The molecular weights of the visible markers of the protein ladder are highlighted with blue arrows. (**B**) Western blot of clones expressing GBS III CPS. Clones were selected following initial screening by dot blot. (**C**) Schematic of each GBS III CPS expressing clone’s Level 3 plasmid. The promoters (P), RBSs (R) and terminators (T) are marked alongside their relative strength, as per Taylor et al., where 1 is the strongest and 6 is the weakest. Figure created with Biorender.com (URL accessed 12/11/2024).

**Figure 3 vaccines-13-00279-f003:**
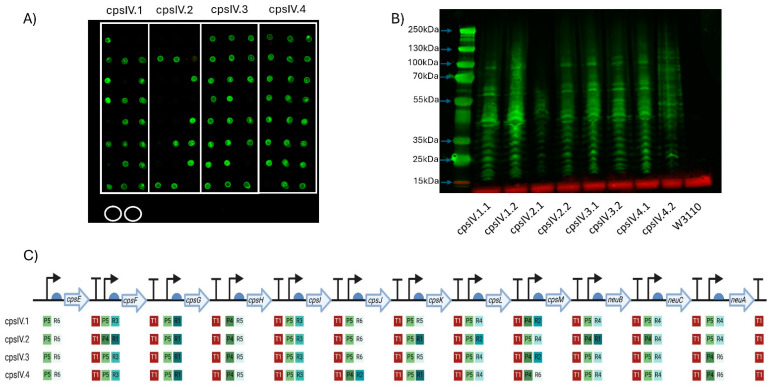
Expression of GBS serotype IV CPS in W3110. (**A**) Dot blot for screening expression of CPS IV. Clones of four plasmids (cpsIV.1-cpsIV.4) were screened, with colonies for each demarcated by the white boxes. Circled in white are biological replicates of W3110 only, used as a negative control. Detection was carried out using anti-GBS IV antisera. The molecular weights of the visible markers of the protein ladder are highlighted with blue arrows. (**B**) Western blot of clones expressing GBS IV CPS. Clones were selected following initial screening by dot blot. (**C**) Schematic of each GBS IV CPS expressing clone’s Level 3 plasmid. The promoters (P), RBSs (R) and terminators (T) are marked alongside their relative strength, as per Taylor et al., where 1 is the strongest and 6 is the weakest. Figure created with Biorender.com (URL accessed 12 November 2024).

**Figure 4 vaccines-13-00279-f004:**
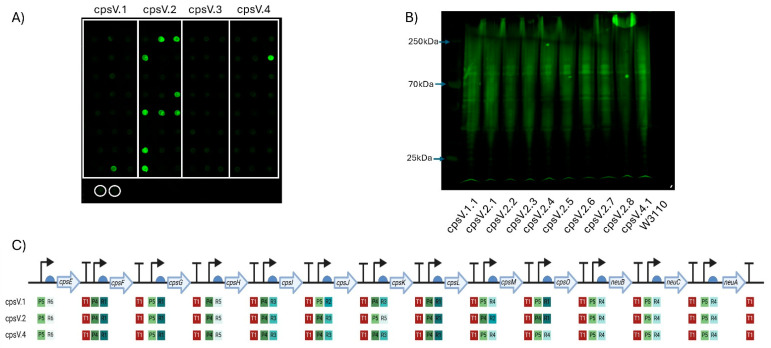
Expression of GBS serotype V CPS in W3110. (**A**) Dot blot for screening expression of CPS V. Clones of four plasmids (cpsV.1–cpsV.4) were screened, with colonies for each demarcated by the white boxes. Circled in white are biological replicates of W3110 only, used as a negative control. Detection was carried out using anti-GBS V antisera. (**B**) Western blot of clones expressing GBS V CPS glycan. Clones were selected following initial screening by dot blot. The molecular weights of the visible markers of the protein ladder are highlighted with blue arrows. (**C**) Schematic of each GBS V CPS expressing clone’s Level 3 plasmid. The promoters (P), RBSs (R) and terminators (T) are marked alongside their relative strength, as per Taylor et al., where 1 is the strongest and 6 is the weakest. Figure created with Biorender.com (URL accessed 12 November 2024).

**Table 1 vaccines-13-00279-t001:** Strains used in this study.

Strain	Genotype	Reference
*E. coli* DH10β	F^–^ *endA1 deoR*^+^ *recA1 galE15 galK16 nupG rpsL* Δ(*lac*)*X74* φ80*lacZΔM15 araD139* Δ(*ara,leu*)*7697 mcrA* Δ(*mrr-hsdRMS-mcrBC*) Str^R^ λ^–^	New England Biolabs
*E. coli* W3110	F- *mcrA mcrB* In(*rrnD-rrnE*)1	[27]

**Table 2 vaccines-13-00279-t002:** Plasmids used in this study.

Plasmid	Description	Source
pStA0	Start-Stop Level 0 vector	[9]
pGT326	pStA0 carrying promoter P4	[9]
pGT327	pStA0 carrying promoter P5	[9]
pGT332	pStA0 carrying RBS R1	[9]
pGT331	pStA0 carrying RBS R2	[9]
pGT330	pStA0 carrying RBS R3	[9]
pGT333	pStA0 carrying RBS R4	[9]
pGT335	pStA0 carrying RBS R5	[9]
pGT334	pStA0 carrying RBS R6	[9]
pGT337	pStA0 carrying terminator T1	[9]
pStA1AB	Start-Stop Assembly Level 1 vector (A and B fusion sites)	[9]
pStA1BC	Start-Stop Assembly Level 1 vector (B and C fusion sites)	[9]
pStA1CD	Start-Stop Assembly Level 1 vector (C and D fusion sites)	[9]
pStA1DE	Start-Stop Assembly Level 1 vector (D and E fusion sites)	[9]
pStA1CZ	Start-Stop Assembly Level 1 vector (C and Z fusion sites)	[9]
pStA1DZ	Start-Stop Assembly Level 1 vector (D and Z fusion sites)	[9]
pStA1EZ	Start-Stop Assembly Level 1 vector (E and Z fusion sites)	[9]
pStA212	Start-Stop Assembly Level 2 vector (1 and 2 fusion sites)	[9]
pStA223	Start-Stop Assembly Level 2 vector (2 and 3 fusion sites)	[9]
pStA234	Start-Stop Assembly Level 2 vector (3 and 4 fusion sites)	[9]
pStA314	Start-Stop Assembly Level 3 vector (1 and 4 fusion sites)	[9]
pCPSIII.1	GBSIII Level 3 clone	This study
pCPSIII.2	GBSIII Level 3 clone	This study
pCPSIII.3	GBSIII Level 3 clone	This study
pCPSIII.4	GBSIII Level 3 clone	This study
pCPSIII.5	GBSIII Level 3 clone	This study
pCPSIII.6	GBSIII Level 3 clone	This study
pCPSIII.7	GBSIII Level 3 clone	This study
pCPSIII.8	GBSIII Level 3 clone	This study
pCPSIV.1	GBSIV Level 3 clone	This study
pCPSIV.2	GBSIV Level 3 clone	This study
pCPSIV.3	GBSIV Level 3 clone	This study
pCPSIV.4	GBSIV Level 3 clone	This study
pCPSV.1	GBSV Level 3 clone	This study
pCPSV.2	GBSV Level 3 clone	This study
pCPSV.3	GBSV Level 3 clone	This study
pCPSV.4	GBSV Level 3 clone	This study
pCPSEFGHI.1	GBSIII Level 2 clone	This study
pCPSEFGHI.2	GBSIII Level 2 clone	This study

**Table 3 vaccines-13-00279-t003:** Oligonucleotides used in this study.

Oligonucleotide	Sequence 5′ to 3′	Description
pStA0_F	GGGGAAACGCCTGGTATCT	Start-Stop Level 0 plasmid forward sequencing primer
pStA0_R	AGCAAAAACAGGAAGGCAAA	Start-Stop Level 0 plasmid reverse sequencing primer
pStA1_F	GTTGAGGACCCGGCTAGG	Start-Stop Level 1 plasmid forward sequencing primer

## Data Availability

Data supporting this study will be made available by the authors upon reasonable request.

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
