# Peer review of "Modular Combinatorial DNA Assembly of Group B Streptococcus Capsular Polysaccharide Biosynthesis Pathways to Expediate the Production of Novel Glycoconjugate Vaccines"

_vaccines, 2025, doi:10.3390/vaccines13030279_

Round 1

Reviewer 1 Report

Comments and Suggestions for Authors

The authors design an innovative vaccine approach for heterologous expression of Group B Streptococcus agalactiae capsular polysaccharides in E. coli using modular combinations of biosynthetic genes. Although this approach has been used previously, this study represents an extension and a thoroughly supported proof-of-principal for 3 glycans representing some of the major disease-associated serotypes of GBS. This work provides a starting point for some of the components of a future GBS vaccine and a model for this approach with other GBS polysaccharides and other microbial systems. The experimental design is reasonable and the results support the interpretations and conclusions.

I think the main relative limitation of the study, which does not constitute a significant weakness, is that evaluation of structurally successful glycan production is entirely based on in vitro dot and Western immunoblotting. These techiques are great for screening and represent important and necessary first steps, but do not guarantee vaccine efficacy. Demonstration of functional utility will require other experiments such as demonstration of in vitro opsonophagocytosis and perhap ultimately protective efficacy in infection models. The authors accurately cite the use of these methods in other related publications, which serve as a rationale and justification for this study, but are choosing to leave this sort of work for the future. This point should be acknowledged, but does not require further experimentation for this report.

Author Response

Comment 1: The authors design an innovative vaccine approach for heterologous expression of Group B Streptococcus agalactiae capsular polysaccharides in E. coli using modular combinations of biosynthetic genes. Although this approach has been used previously, this study represents an extension and a thoroughly supported proof-of-principal for 3 glycans representing some of the major disease-associated serotypes of GBS. This work provides a starting point for some of the components of a future GBS vaccine and a model for this approach with other GBS polysaccharides and other microbial systems. The experimental design is reasonable and the results support the interpretations and conclusions.

Response 1: Our thanks to the reviewer for their time in reviewing our study and for their positive comments on our work.

Comment 2: I think the main relative limitation of the study, which does not constitute a significant weakness, is that evaluation of structurally successful glycan production is entirely based on in vitro dot and Western immunoblotting. These techiques are great for screening and represent important and necessary first steps, but do not guarantee vaccine efficacy. Demonstration of functional utility will require other experiments such as demonstration of in vitro opsonophagocytosis and perhap ultimately protective efficacy in infection models. The authors accurately cite the use of these methods in other related publications, which serve as a rationale and justification for this study, but are choosing to leave this sort of work for the future. This point should be acknowledged, but does not require further experimentation for this report.

Response 2: We thank the reviewer for their time reviewing the manuscript and their positive comments. We agree that a point acknowledging further work is a worthy inclusion and have added the following to the discussion:

Lines 406 – 409

“Validation of a functional immune response to these glycans and any glycoconjugates that include them is essential, therefore future study should include a combination of in vitro assays such as opsonophagocytic killing (OPK), and a challenge model of infection following vaccination.”

Reviewer 2 Report

Comments and Suggestions for Authors

In this manuscript, Harrison et al, addressed the “Modular combinatorial DNA assembly of Group B Streptococcus capsular polysaccharide biosynthesis pathways to expediate the production of novel glycoconjugate vaccines ”. Having examined the manuscript, I note that though it discusses interesting observations, to be considered for MDPI Vaccines, the following are some of the comments that the authors might find useful for future submission. This manuscript is well-structured and delivers insightful information regarding novel glycoconjugate vaccine strategy for controlling group B Streptococcus infections. This type of studies are extremely valuable for the scientific community at a global level.

Reviewer Comments

1.     The authors used Sanger sequencing and Western blotting to for validating glycan structures. If feasible, I would recommend additional methods such as NMR or  Mass spectroscopy for detailed structural characterization of capsular polysaccharides.

2.     The study would be more imapactful, if the authors can include in vitro functional assays  and in vivo safety and preclinical immunization studies.

3.     Some lines in the manuscript is formatting inconsistency. For instance, line 65 to 72. Authors need to correct these lines to ensure uniformity throughout the manuscript.

Author Response

Comment 1: The authors used Sanger sequencing and Western blotting to for validating glycan structures. If feasible, I would recommend additional methods such as NMR or  Mass spectroscopy for detailed structural characterization of capsular polysaccharides.

Response 1: We agree with the reviewer that NMR or Mass Spectroscopy would be valuable validations to our findings. We note that LC-MS/MS mass spectrometry has been performed in our previous publication to characterise the C. jejuni N-linked glycan (Passmore et al., reference 13), however, these methods are poorly suited to characterisation of polymerised glycans such as the GBS capsular polysaccharides. NMR structural characterisation of polymerised glycans is a technically challenging and specialised discipline, which unfortunately we do not have routine access to. We note that the recombinantly synthesised GBS CPS exhibited fidelity to the native structures in the Duke et al study (reference 25), but recognise that further structural characterisation should be performed as part of any future studies.

Comment 2: The study would be more imapactful, if the authors can include in vitro functional assays  and in vivo safety and preclinical immunization studies.

Response 2: We completely agree with this comment, and are currently seeking funding opportunities to continue the work. To acknowledge this point, we have modified the discussion (lines 406-409) to reflect the accuracy of this suggestion and say that these experiments should form part of the future work to test the efficacy of glycans into vaccine candidates:

“Validation of a functional immune response to these glycans and any glycoconjugates that include them is essential, therefore future study should include a combination of in vitro assays such as opsonophagocytic killing (OPK), and a challenge model of infection following vaccination.”

Comment 3: Some lines in the manuscript is formatting inconsistency. For instance, line 65 to 72. Authors need to correct these lines to ensure uniformity throughout the manuscript.

Response 3: We have fixed this inconsistency, and thank the reviewer for spotting the error.

Reviewer 3 Report

Comments and Suggestions for Authors

vaccines-3411837

Modular combinatorial DNA assembly of Group B Streptococcus capsular polysaccharide biosynthesis pathways to expediate the production of novel glycoconjugate vaccines

The work is devoted to the development of approaches for heterologous expression of polysaccharide antigens of group B streptococci, intended for the construction of a multivalent vaccine. The study demonstrates advanced molecular biology and bioengineering approaches and can be recommended for publication in journals of this profile. However, in my opinion, the paper is not quite suitable for Vaccines because it is limited to the expression of target antigens in a heterologous producer (E. coli). For publication in Vaccines, it lacks parts devoted to the development and production of antigenic material, its purification and toxicity assessment, characterization of the structure, multivalent composition, study of immunogenicity and protectivity.

The following points should also be addressed:

L 12 and through the text. ‘polysaccharide (CPS) glycan’ seems a tautology

L 187. Fig 1 Correct a typo in the title

L 135, 188. Please make it clear why text and Fig 1 footnote mentions level 3 plasmids but these plasmids indicated as level 2 in the figure 1. Is it possible to demonstrate a principle scheme of constructs from level 1 to level 3?

L191, 195 Decode CDS and RBC at first usage.

Fig2-4. Specify MW markers in WBs. What kind of antigens in cellular lysates are visualized as multiple bands? Pure CPSs, homo/heterologous GBS lysates controls or usage of specific monoclonal antibodies are desired for antigen identification in WB.

It remains unclear how the issue of bioconjugation (non-chemical conjugation) of polysaccharides with protein antigens was resolved

The chemical structures of GBS-CPS-III-V monomers should be shown to illustrate antisera cross-reactivity.

Round 2

Reviewer 3 Report

Comments and Suggestions for Authors

I admit that the authors have proposed an innovative approach for the production of recombinant group B streptococcal capsular polysaccharide in E. coli. Despite the fact that minor corrections were made in the manuscript, the main issues regarding the characterization of the antigen, its bioconjugate form, immunogenic and protective properties are not covered in the paper. Therefore, in my opinion, the article does not correspond to the profile of Vaccines and is more suitable for Applied Microbiology, Bioengineering, Current Issues in Molecular Biology, or Polysaccharides.
